# Impact of a Multidisciplinary Approach on Cardiometabolic Risk Reduction in a Multiracial Cohort of Adults: A 1-Year Pilot Study

**DOI:** 10.3390/nu14163391

**Published:** 2022-08-18

**Authors:** Ramfis Nieto-Martínez, Andrés Velásquez-Rodríguez, Claudia Neira, Xichen Mou, Andres Neira, Gabriela Garcia, Pedro Velásquez-Rodríguez, Marian Levy, Jeffrey I. Mechanick, Pedro A. Velásquez-Mieyer

**Affiliations:** 1Lifedoc Health, Memphis, TN 38119, USA; 2Departments of Global Health and Population and Epidemiology, Harvard TH Chan School of Public Health, Harvard University, Boston, MA 02115, USA; 3Foundation for Clinic, Public Health, Epidemiology Research of Venezuela (FISPEVEN INC), Caracas 1010, Venezuela; 4LifeDOC Research, Memphis, TN 38119, USA; 5School of Public Health, The University of Memphis, Memphis, TN 38152, USA; 6DarSalud Management, Memphis, TN 38115, USA; 7The Marie-Josée and Henry R. Kravis Center for Cardiovascular Health at Mount Sinai Heart, Division of Endocrinology, Diabetes and Bone Disease, Icahn School of Medicine at Mount Sinai, New York, NY 10029, USA

**Keywords:** multidisciplinary, obesity, risk factors, body mass index, cardiometabolic

## Abstract

Evidence examining specific effects of a multidisciplinary team (MDT) on cardiometabolic risk factors (CMRFs) among multi-ethnic patients in real-world clinical settings is lacking. This one-year retrospective chart review (2018) analyzed 598 adults (African American 59%, Hispanic 35%, and Caucasian 6%) with mean age of 43.8 ± 14.0 years. Qualifying patients with primary inclusion criteria of having body mass indices and blood pressure (BP) measurements in the first and last quarter of the study period were treated under an MDT protocol and compared to those qualifying for MDT but treated solely by a primary care provider (PCP). MDT included endocrinologist-directed visits, lifestyle counseling, and shared medical appointments. MDT patients experienced a greater reduction (β; 95% CI) in weight (−4.29 kg; −7.62, −0.97), BMI (−1.43 kg/m^2^; −2.68, −0.18), systolic BP (−2.18 mmHg; −4.09, −0.26), and diastolic BP (−1.97 mmHg; −3.34, −0.60). Additionally, MDT patients had 77%, 83%, and 59% higher odds of reducing ≥5% of initial weight, 1 BMI point, and ≥2 mmHg DBP, respectively. Improvements in hemoglobin A1C measurements were observed in the MDT group (insufficient data to compare with the PCP group). Compared to PCP only, MDT co-management improves CMRF related to adiposity and hypertension in a multiethnic adult cohort in real-world clinical settings. Patient access to best practices in cardiometabolic care is a priority, including the incorporation of culturally adapted evidence-based recommendations translated within a multi-disciplinary infrastructure, where competing co-morbidities are better managed, and associated research and education programs can promote operational sustainability.

## 1. Introduction

The prevention and treatment of chronic cardiovascular disease (CVD) by focusing on metabolic drivers, such as abnormal adiposity, dysglycemia, hypertension (HTN), and dyslipidemia, require a complex cardiometabolic approach. This mandates the concurrent consideration of both biological and social determinants of health that impact patient care and outcomes [1]. Healthcare delivered by a team following a systematically designed set of multidisciplinary protocols can increase the effectiveness of interventions to mitigate cardiometabolic risk. This multidisciplinary team approach (MDT) has demonstrated effectiveness in controlling weight and associated complications within the controlled conditions of clinical trials. However, implementation in routine, real-world clinical settings with diverse populations based on ethnocultural factors has been limited. This is likely due to suboptimal chronic disease modeling (step 1 below), infrastructure building (step 2 below), sustainability, and effectiveness. Notably, adaptations of lifestyle medicine interventions for diabetes prevention based on the U.S. Diabetes Prevention Program and the Finnish Diabetes Prevention Study have been employed in a variety of clinical settings, but still, these results have been significantly less effective than the original report [2].

Chronic disease modeling is a critical first step to optimize and implement preventive care. To promote early intervention in patients and enhance sensitivity for detecting subjects affected by abnormal adiposity, the American Association of Clinical Endocrinology (AACE) proposed that obesity should be considered not solely by anthropometrics (e.g., ethnicity-adjusted body mass index (BMI) and waist circumference), but by prioritizing weight-related complications. Furthermore, to incorporate new scientific relationships among ectopic fat, inflammation, insulin resistance, and cardiometabolic health, AACE devised the Adiposity-Based Chronic Disease (ABCD) model (stage 1 = risk; stage 2 = predisease; stage 3 = disease; and stage 4 = complications), within which obesity is part of stage 3. Subsequently, to reconcile insulin resistance, prediabetes, type 2 diabetes (T2D), and vascular complications in the same four-stage preventive care strategy, AACE devised the Dysglycemia-Based Chronic Disease (DBCD) model, of which T2D is also stage 3 [3,4]. Unlike the BMI-centric approach to obesity, the complication-centric approach to intervention goals considers abnormal fat distribution and function, as well as ethnocultural, social determinants of health. In a more recent advancement, an umbrella three-dimensional (stage, driver, and social) Cardiometabolic-Based Chronic Disease (CMBCD) model was assembled to provide more breadth and robustness when deriving individual care plans [4,5]. In these driver-based chronic disease models, the consideration of primary drivers (genetics, environment, and behavior creating a unique personalized lifestyle), and well as the secondary/metabolic drivers mentioned above can fashion an approach that mitigates progression to CVD [4].

Infrastructure building is the second step to optimize and implement preventive care. Lifedoc Health (LDH) is a multi-disciplinary and data-driven healthcare organization committed to preventing two of the leading drivers of CVD in the CMBCD model—dysglycemia (specifically T2D) and abnormal adiposity (specifically obesity)—by increasing accessibility to care to minority populations that are underserved through an integrated and standardized outcome-oriented model. This broader approach is necessary for successful population health in the cardiometabolic space. The LDH programs have received state and National Committee for Quality Assurance recognition and accreditation. The LDH clinical model combines primary and specialty care, acute and chronic care, as well as care coordination, pharmacy, patient education, and lifestyle counseling into a unified dynamic approach. Providers undergo protocol training and reinforcement; primary care providers (PCPs) are coached for the early enrollment of patients with diverse ethnic backgrounds, with or without CMBCD risk factors for the MDT co-management of increased weight, high blood pressure, dyslipidemia, and dysglycemia.

Several obstacles may limit the effectiveness of MDT co-management in these patients, including (a) patient and provider perceptions or stigmatization of obesity [6,7], (b) time constraints of providers and limited training to manage obesity and related complications, (c) competing priorities for referral of those with multiple chronic conditions, (d) the presence of numerous social determinants of health (e.g., limited access to preventive care, and job, transportation, and housing insecurity), and (e) patient inertia (e.g., due limited health literacy about their family’s health). The purpose of this pilot study is to demonstrate the effectiveness of implementing the LDH model. This will be accomplished by evaluating the specific impact of a critical aspect of LHD: the effect of MDT on measured markers of CMBCD progression in a multi-racial population with routine, real-world patient encounters.

## 2. Materials and Methods

### 2.1. Study Design and Population

This is a retrospective chart review based on a 1-year analysis (2018) of longitudinal structured data. Data were collected in an electronic health record system from a cohort of 43,079 patients evaluated at LDH over a 15-year period and comprised 83% minorities, including 58% Hispanic (H) and 25% African American (AA). Patients were 56.6% female, with 75% of healthcare coverage coming via government insurance, 10% from commercial insurance, and 15% cash/uninsured. From this, a total of 3596 patients ≥ 18 years of age were evaluated at LDH in 2018 (±6 weeks). Of these, a total of 1112 patients (30.9%) meeting preliminary inclusion criteria, namely BMI and blood pressure averages documented in both the first quarter (Q1) and last quarter (Q4) of the period, were randomly selected. Additional inclusion criteria were included to further refine the patient sample by having the following diagnoses: (1) overweight (OW)/obesity (OB) (via ICD-10 or BMI ≥ 25), (2) HTN stage 1 or 2 (via ICD-10, systolic blood pressure (SBP) between 130–139 or ≥140 mmHg, respectively and/or diastolic blood pressure (DBP) between 80–89 or ≥90 mmHg, respectively), and/or (3) prediabetes/T2D (via ICD-10 or hemoglobin A1c (A1C) ≥ 5.7% (38 mmol/mol) and 6.5% (48 mmol/mol), respectively). Dyslipidemia was not included as an activation diagnosis because a fasting lipid profile was not available for all patients. From this subgroup, 598 patients (53.8%) were ultimately included in the analysis. The exclusion criterion was an inadequate visit profile for either intervention group (Figure 1). The demography of the sample selected was 65% government insurance, 18% commercial insurance, and 17% cash/uninsured, and AA 58.9%, H 35.3%, and Caucasian (C) 5.9%.

### 2.2. Study Variables

The primary exposure in this pilot study was type of care (MDT vs. PCP only). Primary outcomes were 1-year changes in weight and BMI between Q1 and Q4, while secondary outcomes included 1-year changes in SBP, DBP, and number of cardiometabolic risk factors (CMRFs) in the same period. Weight was measured with light clothing and without shoes, using a calibrated scale (Digital Platform Scale Pro Plus 2101KL, Health-o-meter^®^, McCook, IL, USA). Height was measured using a 264 Digital Stationary Stadiometer (Seca^®^, Chino, CA, USA). Blood pressure was measured in the right arm, using the appropriate cuff size, in a sitting position, with a validated Lumeon aneroid sphygmomanometer (McKesson^®^, Irving, TX, USA). The A1C was measured using both in-house rapid testing (Alere Afinion 2 Point of Care Analyzer, Abbott^®^, Lake Forest, IL, USA) and laboratory test by immunoassay in a certified laboratory [8].

### 2.3. Definitions

Ethnocultural classifiers were self-reported as either AA, H, or C. Obesity was defined as a BMI ≥ 30 kg/m^2^ and overweight as BMI 25–29.9 kg/m^2^ in this multiethnic population without Asian representation (in whom BMI cutoffs are lower) [9]. HTN was defined as SBP ≥ 140 mmHg, DBP ≥ 90 mmHg, or personal history of HTN/use of antihypertensive medication [10]. Prediabetes was defined as either fasting blood glucose between 100–125 mg/dL, 2 h post-load blood glucose level between 140–199 mg/dL or A1C between 5.7% (38 mmol/mol) and 6.4% (46 mmol/mol), inclusive. T2D was defined as either a fasting blood glucose ≥ 126 mg/dL, 2 h post-load blood glucose level ≥ 200 mg/dL, A1C ≥ 6.5% (48 mmol/mol), or personal history of T2D [11].

### 2.4. Study Groups and Interventions

Intervention groups were defined as: (a) MDT—patients with ≥2 endocrinology visits and at least 1 visit with a lifestyle intervention provider in the same period, and (b) PCP—patients evaluated at least one time by PCP with ≤1 endocrinology visit and no visits with any lifestyle intervention provider (nutrition, wellness, or education) within the observation period. MDT included primarily endocrinology-oriented visits coordinated by an endocrinologist and provided by a nurse practitioner, lifestyle counseling provided by an exercise physiologist including physical activity and nutritional counselling, and care coordination. Although cardiac function evaluations and eye exams increased the detection of risk factors/complications, these were not considered as grouping variables in this pilot study design. The number of CMRFs for T2D, HTN, and obesity, as well as the time of exposure to LDH interventions were recorded. All patients included in the analysis were evaluated at an LDH site. To account for variability in time of exposure to the intervention, study patients were classified as either (a) “established patients”, namely those receiving care at LDH between 1 January 2008, and 15 November 2017, or (b) “new patients”, describing attending the center only between 15 November 2017 and 31 March 2018 (end of Q1). To account for any selection bias within the analyzed sample, sociodemographic proportions, insurance coverage, and disease prevalence were compared between included patients (*n* = 598) and those excluded from the analysis (*n* = 515). The results show no significant differences among ethnocultural classifiers, gender, or age between the groups. Additionally, all disease prevalence (except for obesity, 67.0% in included vs. 48.1% in total LDH populations, *p* ≤ 0.001) as well as the proportion of established patients was also similar.

Per LDH’s outcome-oriented protocol (Figure 2), a patient began MDT when ≥1 cardiometabolic conditions (overweight/obesity (OW/OB), prediabetes/T2D, HTN, or A1C > 5.6 (38 mmol/mol)) were detected. The present analysis compared the group of patients co-managed by MDT to a group of patients that should have started MDT but were not and continued only with standard PCP care. Providers in both groups had full access to UpToDate as a protocol to guide their approach. Once MDT co-management was initiated, blood samples were collected following a ≥8 h overnight fast and 75 g oral glucose tolerance test with serum glucose and insulin samples at 0, 30, 60, 90, and 120 min drawn [12]. Patients underwent treatment based on a previously established protocolized drug curriculum used at LDH, arrived at based on not only individual clinical conditions but also with a large experience with insurance coverage specifications for the LDH patient population. Specifically, those with OW/OB were treated with topiramate (25 mg po BID) and a low dose of phentermine (18.5 mg po QD), unless contraindicated or otherwise not tolerated. Those with insulin resistance were treated with metformin (1000 mg/day initial dose and then up titrated to 2000 mg/day). Patients with T2D were treated with anti-diabetic drugs clinically proven to promote weight loss, primarily metformin and either incretin-based therapies, such as glucagon-like peptide 1-receptor agonists and dipeptidyl-peptidase 4 inhibitors, or sodium-glucose cotransporter-2 inhibitors. LDH protocols are influenced by reimbursement experiences. For example, Qsymia^®^ (Phentermine/Topiramate ER) is not covered by the insurances that serve our population. An alternative was to prescribe them separately; phentermine is paid for by the patient at a low cost and topiramate is covered by insurers. Lifestyle recommendations were provided for nutrition and wellness, with educational visits and nutritional counseling based on caloric/carbohydrate restriction and accompanying physical activity recommendations. In the MDT group, most visits were coordinated, implemented, and performed along with regularly scheduled medical and lifestyle counseling visits. Patients’ individual needs were considered in the implementation of follow-up and engagement protocols in both groups.

### 2.5. Statistical Analysis

Data were analyzed using R (Version 3.6.2). Continuous variables were presented as mean ± SE and baseline differences among two exposure groups were evaluated using a *t*-test. Frequencies were presented as percentages and 95% confidence intervals (CI) and exposure groups compared using a Z-test. Differences in sociodemographic proportions and disease prevalence between those included and excluded from the analysis were evaluated using Pearson’s chi square. Multiple linear regression analysis was used to investigate the relationship between exposure groups and changes for each outcome from Q1 to Q4 for weight, BMI, SBP, DBP, CMRFs, as well as from first to subsequent A1C measurement. Data on pharmacotherapeutic differences were not included as a part of this analysis. Other confounding variables were included in the model and based on the 10% rule [12]. To account for the effect modification, a variable was also included if a significant interaction term was observed in a model consisting of that variable and the exposure group. To explore exposure effects on clinically significant changes in cardiometabolic outcomes, a binary logistic regression analysis (reduction in at least vs. no reduction in weight (5%), BMI (1 kg/m^2^), SBP (2 mmHg), or A1C (0.3%)) was performed. Confounders and effect modifiers were included, following the same rule as the multiple linear regression. A *p*-value of <0.05 was considered statistically significant. The patients consented to the use of the identified data for research purposes.

## 3. Results

### 3.1. Baseline Subject Characteristics

In total, 598 adults (58.9% AA, 35.3% H, and 5.9% C), with a mean age 43.8 ± 14.0 years were included in the analysis. At baseline, subjects co-managed by MDT were older and heavier than those in the PCP group. The between-group distribution of gender, ethnocultural backgrounds, SBP, DBP, type of patient (established vs. new), and time exposure to interventions were similar. The mean number (±SE) of CMRFs for the total sample was 1.43 ± 1.0, which was higher in the MDT group (1.76 ± 0.93) compared to PCP group (0.72 ± 0.77, *p* ≤ 0.001). Total sample mean A1C was 6.8 (51 mmol/mol) ± 2.0, and higher in MDT group (7.0 ± 2.1) than the PCP group (5.8 ± 1.4, *p* ≤ 0.001). The total sample average number of visits was 10.5, but it was two times higher in the MDT group (mean 12.3) compared to the PCP group (mean = 6.6). In MDT, 30.9% of visits were for lifestyle intervention (Table 1).

### 3.2. Changes in CMRF during the Intervention

At the end of the period, a significant improvement in CMRF was observed across the total sample with changes in mean weight (−0.8 kg, *p* ≤ 0.001), BMI (−0.3 kg/m^2^, *p* ≤ 0.001), SBP (−1.4 mmHg, *p* = 0.01), and DBP (−0.9 mmHg, *p* = 0.01), though changes in the number of CMRF were not significant (−0.04, *p* = 0.16). The positive changes were present in the MDT group: weight (−1.4 kg, *p* ≤ 0.001), BMI (−0.5 kg/m^2^, *p* ≤ 0.001), SBP (−1.5 mmHg, *p* = 0.01), and DBP (−1.1 mmHg, *p* = 0.01), with a reduction in the number of CMRF (−0.1, *p* ≤ 0.001). In contrast, only an increase in the number of CMRF (+0.1, *p* = 0.02) was observed in the PCP group (Table 2). A total of 61 percent of patients (*n* = 364) had a baseline reading for A1C in Q1; among them, 137 (37.6%) had a baseline A1C ≥ 6.5% (48 mmol/mol), and MDT co-management was initiated in 97% of these patients (*n* = 133 of 137). Due to the small number of patients in the PCP group with A1C ≥ 6.5%, between-group changes in A1C were not assessed. As compared with baseline A1C, the MDT group experienced sustained and significant reduction in A1C values, (ΔA1C reading #2 vs. #1 = −0.52% [95% CI −0.89], −0.15; *p* = 0.01; *n* = 132; ΔA1C reading #3 vs. #1 = −0.49% [95% CI −0.91, −0.06]; *p* = 0.03; *n* = 122). Changes between reading #4 and #1 maintained a similar trend, though the significance was lost (−0.32% [95% CI −0.80, −0.15; *p* = 0.18; *n* = 88]).

### 3.3. Factors Related to Changes in Cardiometabolic Risk

The MDT intervention was associated with a higher reduction in adiposity amount (by weight and BMI) and blood pressure. Compared to the PCP group, MDT intervention on average (β, 95% CI) was associated with greater reductions in weight (−4.29 kg, 95% CI −7.62, −0.97), BMI (−1.43 kg/m^2^, 95% CI −2.68, −0.18), SBP (−2.18 mmHg, 95% CI −4.09, −0.26), and DBP (−1.97 mmHg, 95% CI −3.34, −0.60) (Table 3). Binary analysis shows that those co-managed by MDT had 77% higher odds for reducing 5% or more of the initial weight, 83% higher odds of reducing 1 kg/m^2^ of BMI, and 59% higher odds of reducing ≥2 mmHg DBP compared to the PCP group (Table 4). On average, age was associated (β, 95% CI) with greater reductions in weight (−0.09 kg, 95% CI −0.15, −0.03), BMI (−0.03 kg/m^2^, 95% CI −0.06, −0.01), and no reduction in SBP (+0.07 mmHg, 95% CI 0.01, 0.13), while male gender was associated with a higher increase in SBP (3.46 mmHg, 95% CI 1.61, 5.31) and 42% lower odds of reducing ≥2 mmHg than women. Hispanic patients had a higher reduction in SBP (−2.49 mmHg, 95% CI −4.09, −0.26) compared to AA patients. Following the intervention, higher baseline values of weight (only in the continuous outcome analysis), BMI, SBP, DBP, and A1C were associated with a greater reduction in each variable in both continuous and binary outcome analyses.

## 4. Discussion

One of the primary challenges in routine clinical settings is the implementation and adaptation of protocolized interventions. At LDH, patients having at least one cardiometabolic condition (OW/OB, prediabetes/T2D, elevated A1C, or HTN) should begin MDT co-management, though this did not occur in 32% of the study sample. Enabling patients to access best clinical practice in a health system depends on several factors related to the patient, providers, and health system itself.

This study further explored the hypothesis that co-management with MDT improves the CMRF profile compared to management with PCP alone. This pilot study found that MDT care, compared with PCP only, was associated with improvements in adiposity and BP measurements, as well as the number of CMRFs. In addition, MDT patients having an A1C ≥ 6.5% (48 mmol/mol) achieved a sustained and significant improvement compared to baseline readings. In this sample composed largely of ethnic minority patients with OW/OB, increased risk for DBCD, and ≥1 CMRF, the presence of dysglycemia (by A1C) was actually evaluated in only 25% of the PCP patients. This finding is very significant since it suggests that the early screening of dysglycemia may not be as high a priority in the PCP setting, especially when competing with other acute or chronic conditions at the time of the encounter. MDT co-management, on the other hand, may prove timelier and more effective in monitoring and screening the evolution of dysglycemia in at-risk patients in a cardiometabolic multimorbidity model of disease [13]. The notion that T2D can be prevented or delayed if dysglycemia is identified in DBCD stage 2 (predisease) or earlier [14] should be emphasized.

Less weight loss has been reported with lifestyle intervention in AA women (−4.5%) compared to their C and H counterparts (−8.1% and −7.1%, respectively) in the Diabetes Prevention Program [15], as well as male and female AA patients with T2D in the Look AHEAD trial [16]. In the current pilot study, no ethnocultural disparities in weight loss were detected, suggesting a similar benefit across this demographic. A higher reduction in SBP (−2.49 mmHg) was found in H compared to AA, though this was likely not mediated by weight changes. Additionally, the number of CMRFs negatively affected SBP and DBP changes and was associated with an increase of 1.66 and 1.04 mmHg, respectively, though the clinical significance of these findings was not supported by the binary model.

Previous studies have showed the efficacy of an MDT approach. A systematic review and metanalysis including 16 studies compared multidisciplinary collaborative care (defined as care provision by ≥two different care providers) vs. usual care (defined as standard care provided solely by physicians) for patients with uncontrolled diabetes and showed that multidisciplinary collaborative care significantly improved the A1C (mean differences (MD) –0.55%; 95% CI –0.65%, –0.45%; *p* < 0.001) and SBP (MD –4.89 mmHg 95% CI –6.64, –3.13 mm Hg; *p* < 0.001) over 3–12 months [17]. In addition, the perspective of patients and providers on T2D management was reviewed. Although the MDT model improves diabetes treatment outcomes and contributes to complication reduction, trust and synergistic teamwork are important factors in promoting effective care. Patients commended the MDT for improved accessibility. However, a lack of human resources can constrain MDT efficiency, limiting interactions among health care providers, which hampers quality patient care [18].

At the time of this one-year analysis, there was no between-group difference in mean length-of-care, with a total sample average of 61.8 months (Table 1). Those co-managed with MDT averaged twice as many visits in the same period compared to PCP-only patients (12.3 vs. 6.6 visits/y, *p* ≤ 0.001). The findings of this paper support the sustained benefit of a higher number of total visits integrating regular clinical care, care coordination, and lifestyle counseling, which characterize the LDH model. This could explain lower patient attrition and higher levels of engagement, as it presents greater opportunity to reinforce lifestyle education and promote behavioral changes. While benefits of an MDT approach on various CMBCD risks have been reported in randomized, controlled trials, a specialist-based model for patients with severe obesity has been proposed (the “Weight Assessment and Management Clinic”) [19]. In another study, a high-intensity, lifestyle-based treatment program for obesity delivered in an underserved primary care population produced clinically significant weight loss at 2 years [20]. On the other hand, a review of treatment of obesity in primary care practices in the U.S. did not support the exclusive use of low-to-moderate intensity PCP counseling to achieve significant weight loss, though this can be achieved when used in combination with either pharmacotherapy or intensive counseling and meal replacements from a dietician or nurse [21].

The implementation of MDT interventions in clinical settings imposes certain challenges. In this study, MDT provided the opportunity for more interaction with providers, intensive pharmacological management, CMRF screening tools, and lifestyle counseling. At LDH, patients in the MDT group were co-managed by PCPs and most visits were scheduled simultaneously as shared medical appointments where possible to maximize the patient’s convenience. Shared medical appointments involved several care team members, including personnel trained in delivering patient education (lifestyle medicine provider), facilitating patient interaction (medical assistant), and a prescribing provider (endocrinologist or endocrinologist-directed nurse practitioner), initiating and sustaining a comprehensive care plan [22]. Shared medical appointments have been shown to improve A1C and SBP in patients with T2D [23], though there is less evidence of its efficacy in other chronic conditions [24].

The detection and management of CMBCD in a real-world clinical setting is complex as involves multiple drivers, morbidities, and genetic/environmental interactions [13,25]. Protocol initiation promotes the patient’s access to the best healthcare and is subject to both patient-imposed and provider-imposed barriers. In this study, some factors may have limited a patient’s initiation to MDT, such as ethnocultural disparities in weight perception, as well as other social determinants of health, including a lack of transportation or insurance, lower-income, low health literacy, or work-related constraints for appointments, each of which contributes to the proportion of no-shows. Understandably, both time constraints and multi-morbidity can impose barriers to better outcomes in PCP-driven interventions. This ultimately causes PCPs frustration in treating obesity, since limited time for simple lifestyle prescription does not lead to sustainable weight loss or improved management of comorbidities for the majority. Payer impact on patient outcomes should also be recognized, as their preferred fee-for-service reimbursement model contributes to the implementation of a volume-driven practices with more restricted visit durations, making it extremely difficult to cover all aspects of a patient with multiple competing chronic conditions. In this study, patients undergoing MDT co-management were older and heavier, suggesting that weight stigma could also contribute to a lack of provider’s initiation of younger and overweight patients. Clinical inertia—failure of providers to initiate or intensify treatment when indicated—and diagnosis inertia—unawareness/failure to diagnose a condition when present [26]—can also foster lack of protocol initiation. However, contrary to the general idea that providers are chiefly responsible for inertia, all participants in the care delivery experience, including patients, pharmacists, nurses, medical assistants, health authorities, payers, and policy makers, should be considered as playing a role [27]. Rather than an accusatory outlook towards reduced MDT initiation or provider inertia, a clear understanding of its multiple causes and determinants should be captured, and the development of specific integrated strategies in clinical practice should be promoted to overcome these barriers.

There are some limitations of this study. This is a retrospective study and, as such, is liable to residual confounders. Thus, causal inferences cannot be drawn. Since these data were drawn from routine clinical practice, not all were obtained at the same intervals, for which the averages of measurements from Q1 and Q4 were calculated. Similarly, laboratory tests were not available for all participants in Q1 and Q4, eliminating the possibility of analyzing changes in blood glucose and lipid profiles. Only the A1C readings of those patients with baseline measurements ≥ 6.5% were included in T2D range, as those tests are payer-reimbursable. Patients referred to MDT care by PCPs were older and heavier, which may suggest a referral bias, though MDTs remained more effective even after adjusting by baseline age and BMI. This study has the strength of accurately elucidating what occurs in daily clinical practice, including the challenges in analyzing outcomes, particularly in low-income ethnic minority populations, which are otherwise underrepresented in reference studies. The proportion of patients with obesity was higher among those included in the analysis. Considering that its primary outcome is the change in weight and BMI, this is likely associated with a higher level of participation and greater availability of data among obese patients. Therefore, the findings of this pilot study should be suitable to design a larger, more definitive clinical trial.

## 5. Conclusions

There is a need to simplify the translation of evidence-based interventions into daily clinical practice and to ease healthcare accessibility. LDH’s protocolized, integrated, outcome-based clinical MDT model may positively impact the progression of ABCD, DBCD, HTN, and CMBCD, to ultimately reduce the burden of CVD in a low-income, ethnic minority cohort of adults, compared to a PCP-only approach. The results from this analysis can be easily applied to optimize clinical practice, more positively improve health outcomes, and ultimately reduce healthcare system inertia and costs of care in real-world settings. Education and research are critical tools for comprehensive lifestyle medicine programs to optimize sustainability and performance. The significant improvement in the outcomes of the MDT group suggests that other associated factors should be further investigated and identified in large well-designed clinical trials to promote effective strategies during the current epidemic progression of cardiometabolic conditions.

## Figures and Tables

**Figure 1 nutrients-14-03391-f001:**
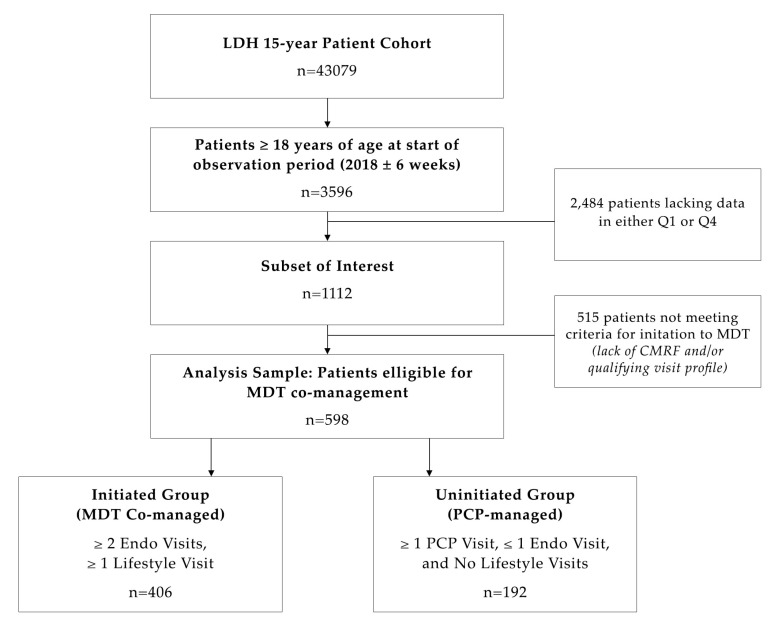
Study sample selection. Abbreviations: CMRF: Cardiometabolic risk factors, Endo: Endocrinology, LDH: Lifedoc Health, MDT: Multidisciplinary team, PCP: Primary Care Provider, Q1: First quarter, Q4: Last quarter.

**Figure 2 nutrients-14-03391-f002:**
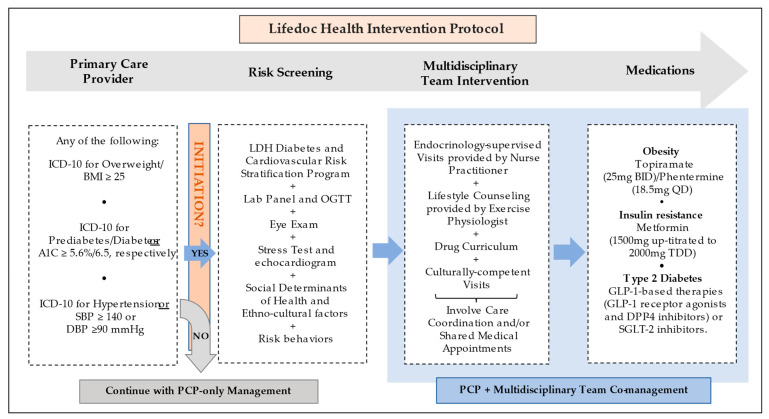
Lifedoc Health intervention protocol. Abbreviations: A1C: hemoglobin A1c, BID: twice a day, BMI: body mass index, DBP: diastolic blood pressure, DPP-4: Dipeptidyl dipeptidase 4, GLP-1: Glucagon-like peptide-1, ICD-10: International Classification of Diseases 10, LDH: Lifedoc Health, OGTT: Oral glucose tolerance test, PCP: Primary Care Provider, QD: Once a day, SBP: systolic blood pressure, SGLT-2: Sodium-Glucose Cotransporter-2, TDD: Total daily dose.

**Table 1 nutrients-14-03391-t001:** Baseline demographics, health care, and cardiometabolic risk factors by type of intervention.

	Primary Care Provider	Multidisciplinary Team	Total	*p*
Demographics				
N (%)	192 (32.1)	406 (67.9)	598 (100)	
Age (years)	39.9 ± 14.0	45.6 ± 13.6	43.8 ± 14.0	≤0.001
Male (%)	21.4 (15.6–27.2)	27.8 (23.5–32.2)	25.8 (22.2–29.3)	0.112
African American (%)	60.4 (53.5–67.3)	58.1 (53.3–62.9)	58.9 (54.9–62.8)	0.658
Hispanic (%)	35.9 (29.2–42.7)	35.0 (30.3–39.6)	35.3 (31.5–39.1)	0.890
Caucasian (%)	3.7 (1.0–6.3)	6.9 (4.4–9.4)	5.9 (4.0–7.7)	0.163
CMRF				
Number of CMRFs	0.7 ± 0.8	1.8 ± 0.9	1.4 ± 1.0	≤0.001
Weight (kg)	84.8 ± 21.5	97.3 ± 26.8	93.3 ± 25.8	≤0.001
BMI (kg/m^2^)	31.7 ± 7.1	36.13 ± 9.3	34.7 ± 8.9	≤0.001
SBP (mmHg)	122.8 ± 14.8	123.64 ± 13.3	123.4 ± 13.8	0.510
DBP (mmHg)	78.16 ± 9.0	77.52 ± 8.3	77.7 ± 8.5	0.387
A1C %, (mmol/mol)	5.8 (40) ±1.4	7.0 (53) ± 2.1	6.8 (51) ± 2.0	≤0.001
Health care indicators				
Number of visits				
Primary Care Provider	6.4 ± 3.0	3.6 ± 4.3	4.5 ± 4.1	≤0.001
Endocrinology	0.2 ± 0.4	4.9 ± 1.7	3.4 ± 2.6	≤0.001
Lifestyle Medicine	0	3.8 ± 4.9	2.6 ± 4.4	≤0.001
Length-of-Care (months)	63.5 ± 40.6	61.0 ± 41.8	61.8 ± 41.4	0.498
Type of patient				
New (%)	9.4 (5.3–13.5)	10.3 (7.4–13.3)	10.0 (7.6–12.4)	0.824
Established (%)	90.6 (86.5–94.7)	89.7 (86.7–92.6)	90.0 (87.6–92.4)	0.824

Frequencies are expressed as percentages and 95% confidence intervals (95% CI) and intervention groups were compared using χ^2^ test. Variables with normal distribution are expressed as mean ± se and intervention groups were compared using a *t*-test. Abbreviations: A1C: hemoglobin A1c, BMI: body mass index, CMRFs: cardiometabolic risk factors (type 2 diabetes, obesity, and hypertension), DBP: diastolic blood pressure, SBP: systolic blood pressure.

**Table 2 nutrients-14-03391-t002:** Changes in weight, blood pressure, and number of cardiometabolic risk factors between quarter 1 and quarter 4 by the type of intervention during the 1 y observational period.

	Primary Care Provider		Multidisciplinary Team		Total	
	Δ Q4–Q1	*p*	Δ Q4–Q1	*p*	Δ Q4–Q1	*p*
Weight (kg)	+0.6 (−0.3, 1.4)	0.19	−1.4 (−2.0, −0.8)	≤0.001	−0.8 (−1.3, −0.3)	≤0.001
BMI (kg/m^2^)	+0.2 (−0.1, 0.6)	0.15	−0.5 (−0.8, −0.3)	≤0.001	−0.3 (−0.5, −0.1)	≤0.001
SBP (mmHg)	−1.2 (−2.9, 0.5)	0.17	−1.5 (−2.6, −0.4)	0.01	−1.4 (−2.3, −0.5)	≤0.001
DBP (mmHg)	−0.4 (−1.4, 0.7)	0.49	−1.1 (−1.9, −0.3)	0.01	−0.9 (−1.5, −0.2)	0.01
Number of CMRF	+0.1 (0.0, 0.2)	0.02	−0.1 (−0.2, −0.04)	≤0.001	−0.04 (−0.1, 0.02)	0.16

Mean changes and 95% confidence intervals (95% CI) are presented. Mean changes between quarter 1 (Q1) and quarter 4 (Q4) for each variable were compared using a paired *t*-test. Δ: mean differences between Q4 and Q1. Abbreviations: BMI: body mass index, CMRF: cardiometabolic risk factors (type 2 diabetes, obesity, and hypertension), DBP: diastolic blood pressure, SBP: systolic blood pressure.

**Table 3 nutrients-14-03391-t003:** Association between demographics and health care variables and changes in cardiometabolic factors (*n* = 598).

CMRF	Explanatory Variable	Linear Regression Analysis β (CI)	
Weight		Weight changes (kg)	*p*
	Type of intervention (MDT vs. PCP)	−4.29 (−7.62, −0.97)	0.011
	Age	−0.09 (−0.15, −0.03)	0.005
	Baseline weight (Q1)	−0.03 (−0.05, −0.00)	0.019
	Number of CMRF	−0.42 (−1.06, −0.22)	0.200
BMI		BMI changes, kg/m^2^	*p*
	Age	−0.03 (−0.06, −0.01)	0.007
	Type of intervention (MDT vs. PCP)	−1.43 (−2.68, −0.18)	0.025
	Baseline BMI (Q1)	−0.04 (−0.07, −0.02)	≤0.001
	# CMRF	−0.08 (−0.32, 0.16)	0.506
SBP		SBP changes, mmHg	*p*
	Age	0.07 (0.01, 0.13)	0.033
	Gender (Male vs. Female)	3.46 (1.61, 5.31)	≤0.001
	Type of intervention (MDT vs. PCP)	−2.18 (−4.09, −0.26)	0.026
	Race (H vs. AA)	−2.49 (−4.25, −0.73)	0.006
	Race (C vs. AA)	−0.83 (−4.22, 2.55)	0.628
	Baseline SBP (Q1)	−0.54 (−0.60, −0.47)	≤0.001
	Number of CMRF	1.66 (0.67, 2.66)	0.001
DBP		DBP changes, mmHg	*p*
	Age	−0.03 (−0.07, 0.02)	0.213
	Type of intervention (MDT vs. PCP)	−1.97 (−3.34, −0.60)	0.005
	Baseline DBP (Q1)	−0.48 (−0.55, −0.41)	≤0.001
	Number of CMRF	1.04 (0.34, 1.74)	0.004
A1C changes		A1C changes, %	*p*
	Age	0.01 (−0.01, 0.02)	0.271
	Gender (Male vs. Female)	−0.28 (−0.65, 0.09)	0.135
	Type of intervention (MDT vs. PCP)	0.32 (−0.25, 0.88)	0.271
	Race (H vs. AA)	0.03 (−0.33, 0.39)	0.858
	Race (C vs. AA)	0.62 (−0.16, 1.39)	0.117
	Baseline A1C (Q1)	−0.24 (−0.32, −0.16)	≤0.001
	Type of patient (New vs. Established)	−0.26 (−0.78, 0.26)	0.327

Data are β coefficient and 95% confidence intervals (95% CI). Linear regression analysis was applied. Abbreviations: A1C: hemoglobin A1c, AA: African American, BMI: body mass index, C: Caucasian, CMRF: cardiometabolic risk factors (type 2 T2D, obesity, and HTN), DBP: diastolic blood pressure, H: Hispanic, MDT: multidisciplinary team, PCP: primary care provider, Q1: first quarter, SBP: systolic blood pressure.

**Table 4 nutrients-14-03391-t004:** Association between demographics and health care variables and changes in cardiometabolic factors (*n* = 598).

CMRF	Explanatory Variable	Binary Logistic Regression Analysis OR (CI)	
Weight		Weight loss at least 5% initial weight (yes, no)	*p*
	Type of intervention (MDT vs. PCP)	1.77 (1.09, 2.94)	0.023
	Baseline weight (Q1)	1.00 (1.00, 1.01)	0.232
BMI		BMI change at least 1-point BMI (yes, no)	*p*
	Type of intervention (MDT vs. PCP)	1.83 (1.16, 2.93)	0.010
	Baseline BMI (Q1)	1.05 (1.02, 1.07)	≤0.001
	Number of CMRF	1.03 (0.83, 1.27)	0.772
SBP		SBP change at least 2 mmHg (yes, no)	*p*
	Age	0.99 (0.98, 1.01)	0.476
	Gender (Male vs. Female)	0.58 (0.38, 0.89)	0.013
	Type of intervention (MDT vs. PCP)	1.36 (0.87, 2.14)	0.175
	Baseline SBP (Q1)	1.08 (1.07, 1.11)	≤0.001
	Number of CMRF	0.81 (0.64, 1.01)	0.067
	Time exposure to intervention (Months)	1.00 (1.00, 1.00)	0.995
DBP		DBP change at least 2 mmHg (yes, no)	*p*
	Age	1.01 (0.99, 1.02)	0.248
	Type of intervention (MDT vs. PCP)	1.59 (1.01, 2.52)	0.046
	Race (H vs. AA)	1.15 (0.76, 1.74)	0.502
	Race (White vs. AA)	0.67 (0.27, 1.57)	0.375
	Baseline DBP (Q1)	1.14 (1.11, 1.18)	≤0.001
	Number of CMRF	0.87 (0.69, 1.09)	0.233
A1C changes		HbA1c change at least 0.3% (yes, no)	*p*
	Type of intervention (MDT vs. PCP)	1.01 (0.37, 3.23)	0.987
	Baseline A1C (Q1)	1.56 (1.35, 1.81)	≤0.001

Data are odd ratio (OR) and 95% confidence intervals (95% CI). Binary logistic regression analysis was applied. Abbreviations: A1C: hemoglobin A1c, AA: African American, BMI: body mass index, C: Caucasian, CMRF: cardiometabolic risk factors (type 2 diabetes, obesity, and hypertension), DBP: diastolic blood pressure, H: Hispanic, MDT: multidisciplinary team, PCP: primary care provider, Q1: first quarter, SBP: systolic blood pressure.

## Data Availability

The data that support the findings of this study are available from the corresponding author, (P.A.V.-M.), upon reasonable request.

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
