# Peer review of "Impact of a Multidisciplinary Approach on Cardiometabolic Risk Reduction in a Multiracial Cohort of Adults: A 1-Year Pilot Study"

_nutrients, 2022, doi:10.3390/nu14163391_

Round 1
Reviewer 1 Report
This is a very nicely designed study investigating the impact of multidisciplinary team expertise on management of complex diseases such as obesity, diabetes and cardiovascular disease. Indeed, this study is very timely, as most of the developed world is experiencing and epidemic of these diseases. I only have a couple of minor suggestions to offer the authors for consideration:
1. Could the authors discuss how many of the participants in the PCP only group were on the GLP-1 and SGLT-2 inhibitors? Does this differ statistically from the number of individuals who were on these drugs who were in the MDT group? Since we know that these inhibitors have a positive effect on weight loss, could the weight loss observed in the MDT group be explained by any such difference?
2. Are there any inferences on the cost (either savings or increase in cost) to the approach using MDT? I understand this may not have been in the scope of this particular paper, but perhaps just an inference...For example, there are some report in the literature in the heart failure field showing incredible cost savings in healthcare (something like 10-fold savings) using a multidisciplinary team in the management of complex diseases (such as HF, as well as the ones discussed here).
Author Response
"Please see the attachment."

Reviewer 2 Report
1. A major limitation of the presented project is a very large discrepancy in the number (n) between the groups. I wonder to what extent the conclusions drawn are reliable?
2. The results were varied by race, and there is no breakdown by gender. This is a very important approach in the context of obesity or insulin resistance and therefore should be supplemented.
3. Were the authors approved by the relevant ethics committee to conduct the research? Lack of information and decision number.
4. Tables were formatted inconsistently with the guidelines of the journal. Moreover, statistically significant differences should be clearly marked with indices (a, b ...).
5. There are no footnotes under the figures.
6. Paragraphs are missing throughout the thesis, and sub-items should be introduced in sections 2 and 3.
7. References section - formatting inconsistent with the journal's guidelines.
Author Response
"Please see the attachment."

Round 2
Reviewer 2 Report
Unfortunately, the authors did not follow the previous comments (the tables are still poorly formatted, explanations are missing under the figures, subchapters 2.1, 2.2, ... were not introduced, the references are not prepared in accordance with the journal guidelines, etc.). The article is not suitable for publication in Nutrients as it stands.